# Junction Piezotronic Transistor Arrays Based on Patterned ZnO Nanowires for High-Resolution Tactile and Photo Mapping

**DOI:** 10.3390/s24154775

**Published:** 2024-07-23

**Authors:** Li Zhang, Runhui Zhou, Wenda Ma, Hui Lu, Yepei Mo, Yi Wang, Rongrong Bao, Caofeng Pan

**Affiliations:** 1CAS Center for Excellence in Nanoscience, Beijing Key Laboratory of Micro-Nano Energy and Sensor, Beijing Institute of Nanoenergy and Nanosystems, Chinese Academy of Sciences, Beijing 100083, Chinapancaofeng@buaa.edu.cn (C.P.); 2School of Chemistry and Chemical Engineering, Guangxi University, Nanning 530004, China; 3Department of Physical Education, Renmin University of China, Beijing 100872, China; 4Institute of Atomic Manufacturing, Beihang University, Beijing 100191, China

**Keywords:** sensor array, piezotronics, transistor

## Abstract

Recently, a great deal of interest has been focused on developing sensors that can measure both pressure and light. However, traditional sensors are difficult to integrate into silicon (Si)-based integrated circuits. Therefore, it is particularly important to design a sensor that operates on a new principle. In this paper, junction piezotronic transistor (JPT) arrays based on zinc oxide (ZnO) nanowire are demonstrated. And the JPT arrays show high spatial resolution pressure and light mapping with 195 dpi. Because ZnO nanowires are arranged vertically above the p-type Si channel’s center of the transistor, the width of the heterojunction depletion region is constricted by the positive piezoelectric potential generated by strained ZnO. In addition, photogenerated charge carriers can be created in the Si channel when JPT is stimulated by light, which increases its electrical conductivity. Consequently, the external pressure and light distribution information can be obtained from the variation in the output current of the device. The prepared JPT arrays can be compatible with Si transistors, which make them highly competitive and make it possible to incorporate both pressure and light sensors into large integrated circuits. This work will contribute to many applications, such as intelligent clothing, human–computer interaction, and electronic skin.

## 1. Introduction

Technological advances have raised the level of requirements for electronic means of detecting external physical and chemical information on intelligent wearable devices, human–computer interaction, and environmental detection [1]. Light and force sensors are essential types of sensors that can be used for visual identification or sensing physical objects [2]. Due to the different mechanisms of traditional light and force sensors, two separate components should be operated individually in integrated circuits. The mechanism principles of force sensors can be classified as resistive [3,4,5,6,7,8,9], capacitive [10,11,12], field-effect transistor [13,14,15,16,17,18,19], piezoelectric [20,21,22], and friction [23,24,25,26], while the mechanism of light sensors is the photoconductive effect of semiconductors. Designing multi-function devices that can detect both light and force simultaneously has become an important problem by combining the operating principles of the two sensors.

Si has been the most widely used material for large-scale and ultra-large-scale integrated circuits as a traditional semiconductor. However, for Si-based devices, it is challenging to interact with mechanical signals. Combining piezoelectric semiconductor materials with Si offers a promising solution for developing Si-based composite optical/force sensors. In 2007 and 2010, Zhong Lin Wang’s research team introduced the basic concepts of piezotronics and piezo-phototronics [27,28], respectively. When strain is applied to a non-centrosymmetric, single-crystal material, the piezoelectric potential generated by the ionic polarization can affect the carrier transport at interfaces and junctions. By combining the advantages of both piezoelectric potential-regulated junction carriers and the broad application range of Si, a JPT structure has been developed on a silicon on insulator (SOI) wafer, which will facilitate dual-mode detection of light and force.

In this work, we report a JPT array sensor based on p-Si/N-ZnO NW structures that maps light and pressure with a spatial resolution of 130 μm. In each pixel, p-Si is used as the channel material to connect the source and drain electrodes with ZnO nanowires grown vertically on the surface. When pressure is applied to the pixel in the sensor, a positive piezoelectric potential is generated at the bottom of the ZnO nanowires, which causes the depletion region of the heterojunction to increase on the Si side of the sensor, thereby weakening the conductivity of the Si [29,30]. The distribution of pressure can be detected by comparing the current change before and after the pressure is applied. When the pixel in the sensor is stimulated by long-wavelength light, the Si absorbs photons to generate electron-hole pairs. Under the electric field induced by the heterojunction, electrons enter the ZnO, while holes remain in the Si, increasing the carrier concentration in the p-Si channel. Therefore, the distribution of light in a small area can be detected by comparing the current change before and after the light stimulation.

## 2. Result and Discussions

Figure 1 illustrates the schematic structure of the JPT sensor array and an electron microscope image of a single pixel. The sensor array is comprised of 18 × 18 JPT units and is located on a 25 × 25 mm SOI wafer. The sensors are composed of two main areas: the functional area, which detects external stimuli, and the external wiring area, which connects the functional area to the test circuit. As shown in Figure 1a–e, the details of the arrays and pixels are represented. In each pixel, a p-type Si channel with a width of 20 μm is used, and ZnO nanowires are grown vertically above the center of the Si channel (Figure 1d,e). When a compressive strain is applied, a positive piezoelectric potential is generated at the bottom of the ZnO nanowire. The piezoelectric potential can regulate the conductivity of the Si channel and detect pressure. In addition, the increase in carrier concentration in the Si channel under the irradiation of light can also be detected by the JPT sensor array. The spatial resolution of the JPT array in the X and Y directions is 130 μm (the resolution can be further improved at the designer’s discretion). The fabrication process of the JPT array is compatible with traditional Si processing. There is a new way of controlling the conductivity of Si, and it can be considered as an extension of the application. Furthermore, the array resolution of the JPT sensors with this structure can be adjusted within a certain range according to the needs, providing a good prospect for future applications, such as high-resolution electronic skin.

As shown in Figure 2a, mapping of the pressure distribution in small areas can be achieved by aligning and applying pressure to the JPT array using a sapphire with designed patterned protrusions and then statistically analyzing the current changes for each pixel. Alternatively, a designed patterned photomask can be overlaid onto the JPT array and irradiated with light to map the optical distribution in small areas based on the current changes for each pixel. The structure of the array sensor is illustrated in Figure 2b, where an etched p-Si array is used as the channel and electrodes made of Ni/Au are placed at the ends to facilitate connection to the testing circuit. SU8 is used to encapsulate a portion of the JPT array after ZnO nanowires are vertically grown in the center of each p-Si channel. The processes for fabricating the devices are shown in Figure 2c. Firstly, the p-type Si channels (80 μm in length and 20 μm in width) were created on a SOI substrate (25 × 25 mm^2^ in size) using an inductively coupled plasma (ICP) etching process. Next, Ni/Au electrodes were deposited using radio frequency (RF) magnetron sputtering as the source and drain. An insulating layer of alumina, 60 nm thick, was deposited between the source and drain electrodes using atomic layer deposition (ALD), achieving insulation between the source and drain electrodes. ZnO nanowires were then grown vertically in the center of the channel to serve as a piezoelectric material for channel regulation. Finally, the functional area of the device was encapsulated with SU8 photoresist. Detailed preparation procedures and optical photographs of the device at each step are provided in the Appendix A. Figure 2d shows the final optical photographs of the JPT array sensor, with the inset demonstrating the optical photographs of a single pixel.

## 3. Individual Device Performance

We designed and prepared a single JPT device (Figure 3) to detect the incident light and the pressure. The preparation process of the single device is shown in Appendix A, with optical photographs of the device in each step. A schematic diagram of the JPT device is shown in Figure 3a,b (inset is an optical image of the device). The reason for using a fabrication process similar to a JPT array to prepare a single JPT sensor is to avoid issues such as adjacent pixel crosstalk that may occur when directly testing a JPT array. The response characterization of the JPT sensor to light was studied by using a 525 nm light-stimulation device. As shown in Figure 3c, the I–V electrical characteristics of individual devices with bias voltages ranging from −1 V to +1 V were examined in the dark and under varying levels of illumination. As the light intensity increases, the current between the source and drain electrode increases due to the increase in carrier concentration in the Si channel. By applying a bias voltage of 1 V, the sensor shows good discrimination capabilities for different light intensities when it is stimulated with periodic light signals (5 s for both light and dark durations, five cycles). A plot of the current variation with light intensity was obtained by extracting the currents from Figure 3d (Figure 3f). The sensor exhibited good reproducibility under a bias voltage of 1 V and periodic light stimulation with an intensity of 8.4 mW/cm^2^ (both light irradiation and interval time of 5 s for 50 cycles). Similar to conventional junction field-effect transistors (JFETs), JPTs can regulate channel carrier transport characterization via piezoelectric potential. When the JPT is under different pressures, a positive piezoelectric potential is generated at the bottom of the ZnO nanowires, which affects the conductivity of the Si channel. The current between the source-drain electrodes of the JPT gradually decreased with increasing pressure as the voltage varied from −1 V to +1 V (Figure 3g). In Figure 3g,i, similar to the curves of the JFET, the output and transfer curves were obtained by polynomial fitting of the extraction currents at different pressures with a source-drain voltage of −1 V. The variation in source-drain current with time is obtained by applying a gradually increasing pressure to the JPT device at a bias voltage of 1V in Figure 3h. As the pressure increases, the device current decreases.

## 4. Working Principle

In order to explain the current changes in JPT devices under light and pressure, schematic band diagrams of piezoelectric and semiconductor optoelectronic theories are presented in Figure 4. In this device (Figure 4a), the C-axis direction of the crystal lattice of the grown ZnO nanowires is oriented vertically upward [31,32]. When the compressive strain is applied, ZnO nanowires are compressively deformed (Figure 4b), resulting in a negative piezoelectric potential on the top of the nanowires and a positive piezoelectric potential on the bottom (contact surface with Si). In the absence of external stimuli, the band diagram of the JPT device is shown in Figure 4c. Due to the difference in Fermi levels, electrons in N-ZnO diffuse into p-Si, and holes in p-Si diffuse into N-ZnO, forming a depletion region at the Si/ZnO interface (the area between the two orange lines in Figure 4c). The built-in electric field direction of the heterojunction is from ZnO to Si. Therefore, the region on the p-Si side can be divided into a channel region and a depletion region. When pressure is applied to the device, a positive piezoelectric potential is generated on the ZnO side in contact with Si, while a negative piezoelectric potential is generated on the opposite side of ZnO. At this point, an analysis of the built-in electric field at the Si/ZnO interface shows that the field strength increases, which enlarges the depletion region in the Si area and reduces the channel region (the area between the two orange lines in Figure 4d expands). This weakens the conductivity of Si. Therefore, pressure can be detected by applying the same voltage to Si and collecting the current changes in Si. Similarly, Figure 4e shows a schematic band diagram of the JPT device before it is stimulated by light. When the device is stimulated by long-wavelength light (photon energy is insufficient to excite electron-hole pairs in ZnO), Si absorbs the photons and generates electron-hole pairs. Subsequently, under the influence of the built-in field, electrons move toward ZnO, and the holes remain in Si (Figure 4f). Consequently, when the channel is stimulated by light (longer wavelength), the hole concentration increases in the channel, which leads to an increase in current. It was not only Si but also ZnO that absorbed photons to produce electron-hole pairs stimulated by shorter wavelength light. A built-in field induces the holes from ZnO to silicon, which increase its conductivity. However, after light stimulation at 365 nm, the photocurrent response appears slowly up and down due to the continuous photoconductivity effect of ZnO (Appendix A).

## 5. High Spatial Resolution Pressure Mapping

As shown in Figure 5a, a testing platform was set up to examine the ability of the JPT array to map the pressure distribution in a small area. The testing platform consisted of three parts: control software, multichannel acquisition system, and piezo nano-positioning stage. Beneath the JPT array sensors was an electronic scale, which displayed the force exerted by a metal rod located on a cantilever onto the JPT array sensors (Figure 5b). Figure 5c illustrates the equivalent circuit schematic of this device. The array sensors were connected to the multichannel acquisition system through a zebra paper and a flexible printed circuit board (PCB) (specific details are shown in Appendix A), and a fixed bias voltage was applied to each pixel to enable sequential current acquisition. Figure 5d shows the initial current of the sensors at 1V bias voltage under indoor lighting conditions. Based on statistical analysis and polynomial fitting, it can be observed that the majority of the current values range from 0.1 μA to 1 μA (Figure 5e). A sapphire substrate with a “30” convex pattern was aligned with the pixels in the JPT array under an optical microscope and secured with tape (Appendix A). By moving the metal rod with a 1D stage, different forces were applied to the JPT array, and the current of each pixel was collected by the multichannel acquisition system. Figure 5f–h display the amount of current reduction in the JPT array at different pressures of 0.8 N, 1.5 N, and 2 N, respectively. It can be observed that the pixel current decreases significantly with increasing pressure. By comparing the heat map of the current reduction in each pixel before and after pressure application to the JPT array, mapping of the pressure distribution in a small area can be achieved.

## 6. High Spatial Resolution Light Mapping

Similar to the pressure test system, the optoelectronic measurement platform for the JPT array consists of three components: control software, a multichannel acquisition system, and a light source (Figure 6a). The intensity of the light source is regulated by applying different voltages from a constant voltage source (Figure 6b). Figure 6c shows the initial current of each pixel after applying a bias voltage of 1 V to the JPT array in a dark environment. According to the statistical analysis and polynomial fit of the initial currents of the 18 × 18 array, most of the currents lie at approximately 2.5E-8 A (Figure 6d). A photomask with a “30” pattern is aligned to the JPT device under an optical microscope (Figure 6e). Figure 6f shows the increase in current for each pixel point after irradiating the JPT array device with different intensities of 122.4 W/cm^2^, 3.1 mW/cm^2^, 8.33 mW/cm^2^, and 17.8 mW/cm^2^, respectively (at 525 nm wavelength). As the light intensity increases, the current of the illuminated pixels in the device circuit also increases. Micro-area light mapping can be achieved with JPT array devices.

## 7. Conclusions

In summary, JPT array sensors can achieve dual-mode detection of both pressure and light. Similar to the principle of JFETs, the detection of pressure is attributed to the positive piezoelectric potential generated by the strain applied to ZnO nanowires, which changes the extent of the depletion region of the ZnO/Si heterojunction. Meanwhile, when photogenerated carriers are generated within the silicon, changes and distributions of light can be detected due to the change in silicon conductivity. Thus, the light and force applied to the JPT can be detected from its source-drain current changes, respectively. Finally, an 18 × 18 JPT array (with a spatial resolution of 130 μm equivalent to a pixel density of 195 dpi) was designed and demonstrated for pressure and light mapping. In contrast to traditional silicon-based semiconductor devices, piezoelectric potential was investigated to regulate silicon conductivity and detect pressure in this manuscript. This allows for the silicon process device to interact directly with mechanical signals. This work extends silicon applications, which have significant implications in multifunctional sensing, e-skin, and wearable devices.

## 8. Experimental Section

### 8.1. Individual JPT Fabricated

The SOI substrates (25 mm × 25 mm) were ultrasonically cleaned with ethanol, acetone, and deionized water. They were dried with compressed air. The JPT overlay marks (Ni/Au, 7 nm/50 nm) were prepared by lithography on the SOI substrates. A protective layer of photoresist was then formed by the second photolithography process. Following that, a p-Si channel (80 μm × 20 μm) was prepared by an ICP etching technique. Then, magnetron-sputtered Ni/Au (7 nm/50 nm) served as the source-drain electrode, which covered a 20 × 20 μm region at both ends of the p-Si. Therefore, the size of the p-Si channel is 40 μm × 20 μm. Additionally, ZnO was sputtered in the center of the channel to form a seed layer with dimensions of 20 μm × 40 μm (approximately 30 nm thick). The ZnO nanowires were then grown on the seed layer through a hydrothermal method (see Hydrothermal Growth in the Appendix A). Finally, the devices were encapsulated by SU8 photoresist (GM 1040, Gersteltec Engineering Solutions). The preparation of a single JPT device and corresponding optical images are shown in the Appendix A.

### 8.2. The Preparation Process for JPT Array Devices

Generally, the preparation process of a JPT array was similar to the preparation of a single JPT array. In the JPT array devices, before the drain electrodes preparation, an Al_2_O_3_ layer (thickness of 60 nm) was added at the source-drain crossover location by atomic layer deposition (ALD) as an insulation layer. Detailed information can be found in the Appendix A on the JPT Array Device Preparation.

### 8.3. Experimental Setup and Measurements

The Keysight B1500A (Keysight Technologies) collected the electrical characteristics of a single JPT device. The strain measurement stage (shown in Figure 6a), which was a combination of two 3D displacement stages (Newport M-462 Series), a cantilever, a pressure bar, a scissor lift stage (MST120, Beijing Feichuang Yida Optoelectronic Technology Co., Ltd.), and an electronic scale (LS-I2000, Shanghai Qingyoutang Industrial Co., Ltd.), can apply pressure and record the magnitude of force. The electrical characteristics of the JPT array devices were measured by a multichannel acquisition system that consisted of Keithley 2612B and 3706A. The optoelectronic performance measurement system consisted of a Maynuo DC Source Meter for powering commercial LEDs and a Thorlabs optical power meter for measuring optical power intensity. The experimental and measuring setup included an optical microscope (Zeiss Observer Z1, ZEISS Group), electron microscopes (SU1510, Hitachi or Nova NanoSEM 450, Thermo Fisher Scientific), a lithography machine (BG-401A, China Electronics Technology Group Corporation), an RF magnetron sputtering (PVD75, Kurt J. Lesker), and an ICP (SI500, SENTECH Instruments).

## Figures and Tables

**Figure 1 sensors-24-04775-f001:**
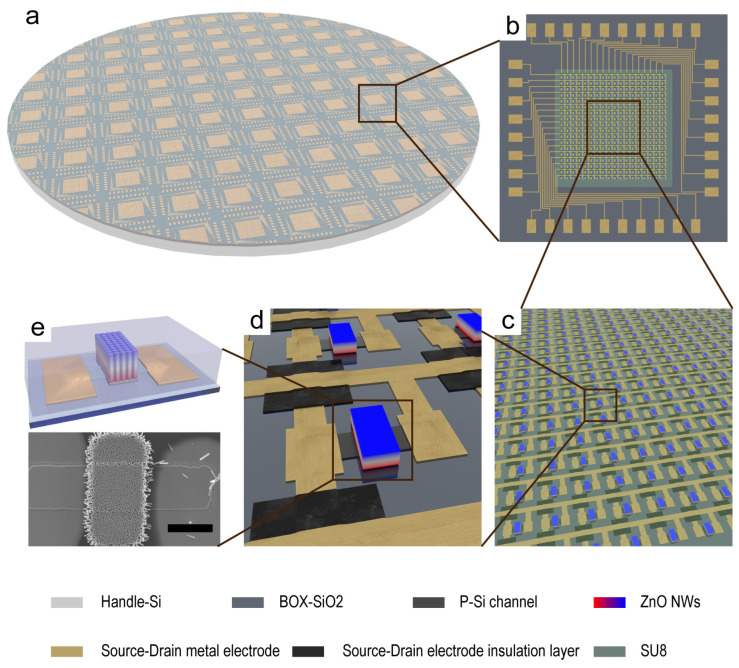
Structure of JPT array. A 25 × 25 mm area (**b**) is cut from a complete SOI wafer (**a**), and the structure of individual pixels in the functional area (**c**) is shown in Figure (**d**,**e**). Scale bar: 20 μm.

**Figure 2 sensors-24-04775-f002:**
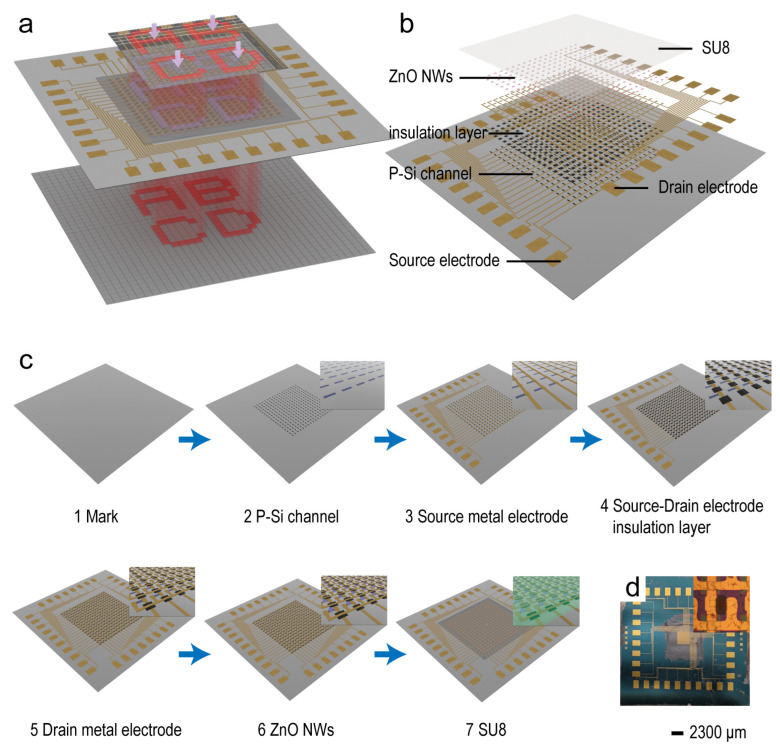
The function, structure, and preparation process of JPT arrays. (**a**) The JPT array enables the mapping of pressure and light. (**b**) Structure of the JPT array. (**c**) Fabrication process of the devices and (**d**) optical photos (inset shows an individual pixel).

**Figure 3 sensors-24-04775-f003:**
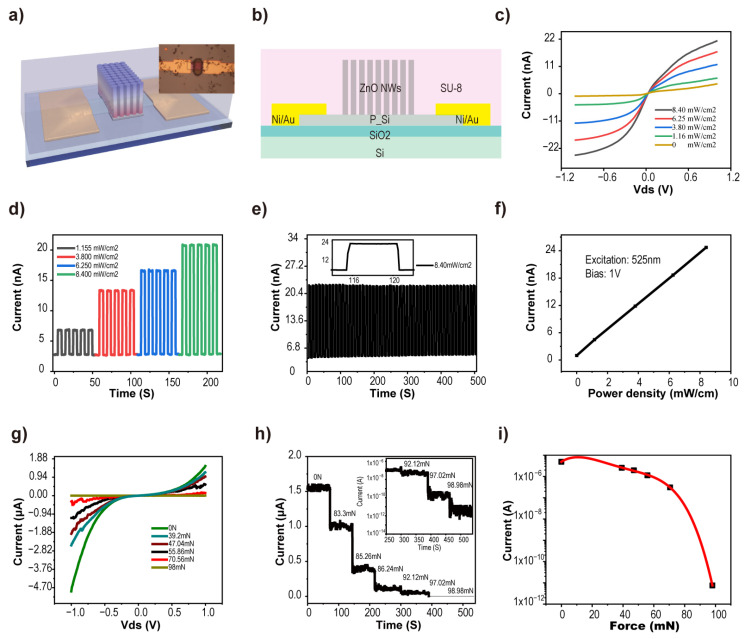
Response of a single JPT for light and pressure. (**a**,**b**) Schematic diagram of the structure of a single device (inset is an optical photograph of the device). (**c**) I-V characteristics of a single JPT in the dark and under various light illumination intensities (525 nm wavelength). Reproducible response of the device under 1 V bias for different illumination intensities (**d**) and stationary illumination intensities (**e**). (**f**) The relationship between illumination intensity and photocurrent of the device at 1 V bias voltage. (**g**) I-V characteristics of a single JPT at different pressures. (**h**) The relation between source leakage current and pressure (at 1 V bias). By extracting the data of the device at −1 V in (**g**), the relationship between current and pressure is plotted in (**i**).

**Figure 4 sensors-24-04775-f004:**
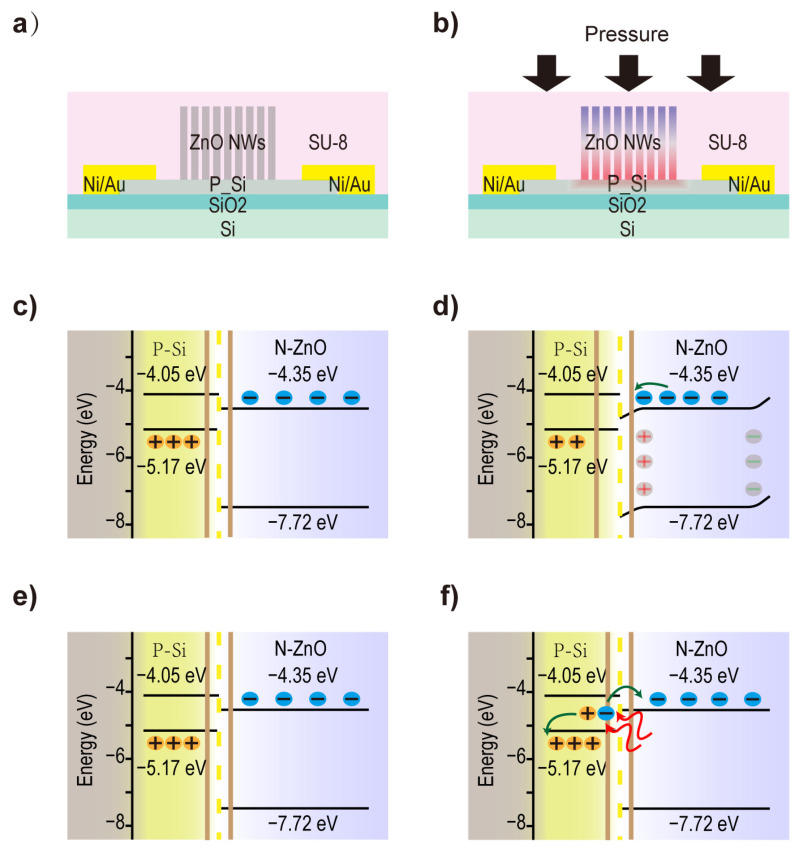
The schematic band diagrams illustrate the physical mechanism of JPT when it is exposed to pressure and light. (**a**) The structure of a JPT. (**b**) Under compressive strain, ZnO nanowires produce positive and negative piezoelectric potentials at the bottom and top, respectively. (**c**) Due to their different Fermi energy levels, Si and ZnO form heterojunctions at the interface. (**d**) When strain is applied, the positive piezoelectric potential generated on the ZnO side increases the depletion zone between silicon and ZnO, which leads to reduced conductivity of silicon. (**e**) Schematic band diagram of JPT before illumination. (**f**) After exposure to longer wavelength light, the electron-hole pairs occur in silicon, where electrons move toward ZnO under the influence of the built-in electric field, leaving holes in silicon to enhance its conductivity.

**Figure 5 sensors-24-04775-f005:**
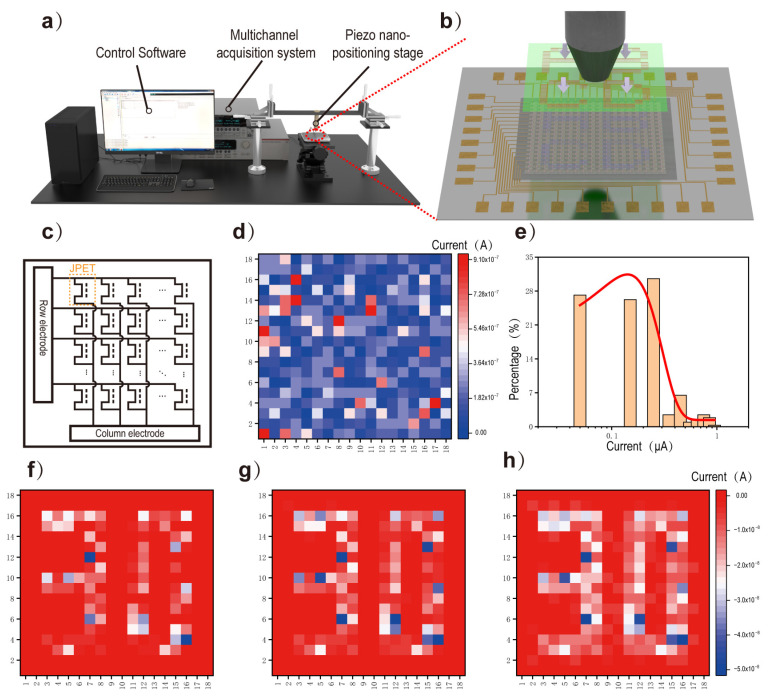
Imaging of pressure distribution. A structure schematic diagram of the test platform (**a**) and the part that applies pressure to the array device (**b**). (**c**) Equivalent circuit diagram of the JPT array. (**d**) Initial current of each pixel of the JPT array at 1 V bias. (**e**) Statistics of the initial currents for each pixel of the JPT array. (**f**–**h**) JPT arrays at 1V bias with increasing pressure gradually increase the variation in current per pixel.

**Figure 6 sensors-24-04775-f006:**
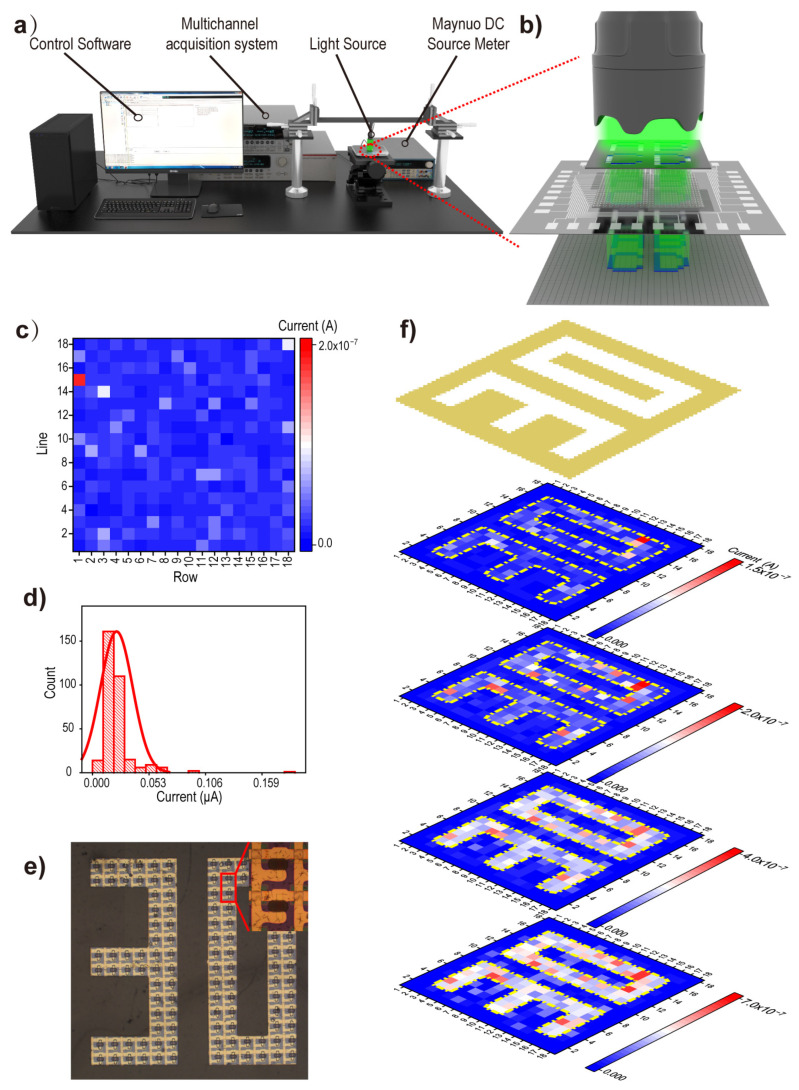
Imaging of light distribution. (**a**) A structure schematic diagram of the test platform and (**b**) the part of the device that is stimulated with light. (**c**) Initial currents of individual pixels with 1 V bias in the dark. (**d**) The initial currents for each pixel of the JPT array are statistically analyzed and fitted. (**e**) Light intensity distribution in tiny areas is simulated by blocking some pixels using a photomask. (**f**) Current increase per pixel after stimulating of the JPT array with a variety of light intensities.

## Data Availability

This study’s data are contained within the article.

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
