# Peer review of "Junction Piezotronic Transistor Arrays Based on Patterned ZnO Nanowires for High-Resolution Tactile and Photo Mapping"

_sensors, 2024, doi:10.3390/s24154775_

Round 1

Reviewer 1 Report

Comments and Suggestions for Authors

Question 1: In the introduction part, the authors should mention the main challenges or limitations faced when integrating light and force sensors into a single device. Explain in detail how piezoelectric semiconductor materials' properties complement Si in sensor applications. How does the spatial resolution of 130 μm compare to other similar sensors in the field? Are there any trade-offs associated with this resolution?  How does the compatibility with traditional Si processing techniques benefit the manufacturing process of these sensors? Are there any challenges associated with this compatibility?

Question 2: How does the single JPT device performance compare to that of the JPT array? Are there any significant differences in sensitivity or accuracy?

Question 3: The authors should clarify how the compressive strain induces a piezoelectric potential and explain how this mechanism is represented in the band diagrams (Figure 4b and 4c)?  Can you provide a more detailed explanation of how compressive strain leads to the creation of a piezoelectric potential, and how this is visually represented in the band diagrams?

Question 4: How do the energy levels of the absorbed photons relate to the different responses of the device under long-wavelength versus short-wavelength light? Is there a specific reason for choosing the 525 nm and 365 nm light sources? Why were the wavelengths of 525 nm and 365 nm chosen for the light stimulation experiments, and how do these wavelengths specifically influence the optoelectronic behavior of the JPT device?

Question 5: Could you expand on the observation of the “slowly up and down” photocurrent response at 365 nm? What are the underlying causes of this behavior? What factors contribute to the slowly varying photocurrent response at 365 nm, and how might this affect the device's performance in practical applications?

Author Response

Point to Point Response to the referees’ reports (comments in black and responses are in blue):

Question 1: In the introduction part, the authors should mention the main challenges or limitations faced when integrating light and force sensors into a single device. Explain in detail how piezoelectric semiconductor materials' properties complement Si in sensor applications. How does the spatial resolution of 130 μm compare to other similar sensors in the field? Are there any trade-offs associated with this resolution?  How does the compatibility with traditional Si processing techniques benefit the manufacturing process of these sensors? Are there any challenges associated with this compatibility?

Response:

Thank the reviewer for the suggestion. In the published literature, there has been no mention of using piezoelectric potential to regulate the conductivity of other channel materials and achieve large-scale arrays. The 130μm spatial resolution is a compromise considering factors such as the success rate of sensor fabrication and operator proficiency. The actual achievable resolution can be adjusted to some extent with advancements in micro-nano fabrication instruments and the required target resolution.

Compared to traditional Si micro-nano processing technology, the JPT sensor fabrication process only additionally includes the hydrothermal growth of ZnO nanowires. During the hydrothermal growth process, there is no highly alkaline or acidic environment, and the sensor can be fabricated under certain protection (e.g., photoresist). If compatibility with Si-based circuits is needed, a series of preliminary experiments should be conducted to determine the corresponding sensor dimensions, ensuring that the range of sensor current variations meets the design requirements of Si-based circuits.

Question 2: How does the single JPT device performance compare to that of the JPT array? Are there any significant differences in sensitivity or accuracy?

Response:

Thank the reviewer for the suggestion. A single JPT device has better performance compared to a single pixel sensor in a JPT array. This is because the current change generated by each pixel sensor in the JPT array when subjected to external stimuli (pressure or light) is influenced by the currents of adjacent or nearby pixels. Specifically, the current of pixels in the array sensor is collected by applying a high level to one row and one column and grounding them, while the other rows and columns are disconnected. By analyzing the equivalent circuit of the array sensor (Figure 5b), it can be observed that the current flows not only through the pixel being read but also forms a loop through other adjacent pixels. This results in a higher initial current in the array pixel compared to a single JPT device and a smaller change in current when subjected to pressure or light stimuli.

Question 3: The authors should clarify how the compressive strain induces a piezoelectric potential and explain how this mechanism is represented in the band diagrams (Figure 4b and 4c)?  Can you provide a more detailed explanation of how compressive strain leads to the creation of a piezoelectric potential, and how this is visually represented in the band diagrams?

Response:

Thank the reviewer for the suggestion. Upon contact between P-Si and N-ZnO, a depletion region forms in the contact area due to carrier diffusion (shown as the area between the two orange lines in Figure 4c). Analyzing this depletion region reveals that the built-in electric field points from ZnO to Si. Thus, the region on the P-Si side can be split into a channel region and a depletion region. When external pressure is applied to ZnO, a positive piezoelectric potential is generated on the ZnO side that is in contact with Si, while a negative piezoelectric potential is generated on the opposite side of ZnO. At this point, analyzing the built-in electric field of the Si/ZnO reveals an increase in its strength, which enlarges the depletion region in the Si area and reduces the channel region (the orange areas in Figure 4d grow). This diminishes the conductivity of Si. Consequently, pressure detection can be achieved by applying the same voltage to Si and monitoring changes in the current within Si.

Question 4: How do the energy levels of the absorbed photons relate to the different responses of the device under long-wavelength versus short-wavelength light? Is there a specific reason for choosing the 525 nm and 365 nm light sources? Why were the wavelengths of 525 nm and 365 nm chosen for the light stimulation experiments, and how do these wavelengths specifically influence the optoelectronic behavior of the JPT device?

Response:

Thank the reviewer for the suggestion. In the experiment, light with wavelengths of 525nm and 365nm was chosen for the following reasons:

1.These two wavelengths are common for LEDs. By fixing the distance between the LED and the sensor and adjusting the LED's driving voltage, different light intensities can be simulated.

2.The energy of 525nm photons can excite electrons from the valence band to the conduction band in silicon (Si) but cannot excite electrons from the valence band to the conduction band in zinc oxide (ZnO). Thus, during photoelectric detection, light can be detected through changes in the current in Si.

3.The energy of 365nm photons can excite electrons from the valence band to the conduction band in both Si and ZnO, thereby increasing the conductivity of the detector. A detailed explanation is provided in question 5.

Question 5: Could you expand on the observation of the “slowly up and down” photocurrent response at 365 nm? What are the underlying causes of this behavior? What factors contribute to the slowly varying photocurrent response at 365 nm, and how might this affect the device's performance in practical applications?

Response:

Thank the reviewer for the suggestion. In the experiment, we used 365 nm light to periodically stimulate a single JPT sensor. It was observed that the current of the sensor increased continuously under a fixed bias as the illumination persisted, and then gradually decreased after the light was removed. This phenomenon is attributed to the persistent photoconductivity effect in ZnO nanowires, which exhibit high defect states due to the hydrothermal growth method. As a result, the photogenerated electron-hole pairs induced by 365 nm light remain in the ZnO nanowires for a prolonged period.

When exposed to 365 nm light, holes in the ZnO nanowires are continuously injected into the Si under the influence of the built-in electric field in the ZnO/Si structure, thereby increasing the conductivity of the Si. After the light source is removed, the electron-hole pairs in the ZnO nanowires do not immediately recombine, and the holes continue to be injected into the Si under the influence of the built-in electric field, resulting in a gradual decrease in current.

Due to these reasons, when using this JPT sensor to detect higher energy light (where the photon energy can excite valence band electrons in ZnO nanowires to the conduction band), there is a noticeable response lag.

Reviewer 2 Report

Comments and Suggestions for Authors

The manuscript reports on the development of junction piezotronic transistor (JPT) arrays based on zinc oxide (ZnO) nanowires. The authors demonstrate that the JPT arrays can achieve high spatial resolution pressure and light mapping with a resolution of 195 dpi. The unique arrangement of ZnO nanowires vertically above the p-type Si channel center enables the constriction of the heterojunction depletion region by the positive piezoelectric potential generated by strained ZnO. Additionally, the photogenerated charge carriers created in the Si channel when stimulated by light increase the electrical conductivity. This allows the external pressure and light distribution information to be obtained from the variation of the output current of the device. The compatibility of the prepared JPT arrays with Si transistors makes them highly competitive and suitable for large integrated circuits. This work is expected to contribute to applications such as intelligent clothing, human-computer interaction, and electronic skin.

However, there are some issues in the manuscript. I recommend this paper for minor revision after addressing the following points:

1. The manuscript demonstrates a resolution of 195 dpi. Is there potential for further increasing the resolution? What are the limiting factors?

2. When using the sensor arrays to detect light and pressure in small areas, is there a solution to avoid current signal crosstalk between different pixels?

3. You mention the potential integration with traditional Si-based circuits. Considering the possible damage to Si-based circuits during sensor fabrication, do you have any recommendations based on your sensor fabrication process?

4. The manuscript mentions that the piezoelectric potential of ZnO affects the bandgap width when detecting pressure. Are there any other piezoelectric materials that could replace ZnO? Please explain the advantages of ZnO compared to other piezoelectric material.

Author Response

Point to Point Response to the referees’ reports (comments in black and responses are in blue):

The manuscript reports on the development of junction piezotronic transistor (JPT) arrays based on zinc oxide (ZnO) nanowires. The authors demonstrate that the JPT arrays can achieve high spatial resolution pressure and light mapping with a resolution of 195 dpi. The unique arrangement of ZnO nanowires vertically above the p-type Si channel center enables the constriction of the heterojunction depletion region by the positive piezoelectric potential generated by strained ZnO. Additionally, the photogenerated charge carriers created in the Si channel when stimulated by light increase the electrical conductivity. This allows the external pressure and light distribution information to be obtained from the variation of the output current of the device. The compatibility of the prepared JPT arrays with Si transistors makes them highly competitive and suitable for large integrated circuits. This work is expected to contribute to applications such as intelligent clothing, human-computer interaction, and electronic skin.

However, there are some issues in the manuscript. I recommend this paper for minor revision after addressing the following points:

Answers:

We like to express our sincere thanks to the referee for her/his great effort to review the manuscript and positive evaluation on our work.

  1. The manuscript demonstrates a resolution of 195 dpi. Is there potential for further increasing the resolution? What are the limiting factors?

Response:

Thank the reviewer for the suggestion. The resolution of 191 dpi is a compromise considering factors such as the success rate of sensor fabrication and operator proficiency. The actual achievable resolution can be adjusted to some extent based on micro-nano processing equipment and the target resolution required in practice.

  1. When using the sensor arrays to detect light and pressure in small areas, is there a solution to avoid current signal crosstalk between different pixels?

Response:

Thank the reviewer for the suggestion. To prevent the problem of current signal crosstalk between pixels in various arrays, it's necessary to ensure that each pixel has unidirectional conductive characteristics. Thus, by incorporating a PN junction in each pixel via micro-nano processing, signal crosstalk can be effectively minimized.

  1. You mention the potential integration with traditional Si-based circuits. Considering the possible damage to Si-based circuits during sensor fabrication, do you have any recommendations based on your sensor fabrication process?

Response:

Thank the reviewer for the suggestion. Compared to traditional Si micro-nano processing technology, the JPT sensor fabrication process only additionally includes the hydrothermal growth of ZnO nanowires. During the hydrothermal growth process, there is no highly alkaline or acidic environment, and the sensor can be fabricated under certain protection (e.g., photoresist). If compatibility with Si-based circuits is needed, a series of preliminary experiments should be conducted to determine the corresponding sensor dimensions, ensuring that the range of sensor current variations meets the design requirements of Si-based circuits.

  1. The manuscript mentions that the piezoelectric potential of ZnO affects the bandgap width when detecting pressure. Are there any other piezoelectric materials that could replace ZnO? Please explain the advantages of ZnO compared to other piezoelectric material.

Response:

Thank the reviewer for the suggestion. ZnO stands out among other piezoelectric materials due to its superior biocompatibility, controllable growth process, ease of meeting growth conditions, lower production costs, high piezoelectric coefficients, and transparency.